# Evaluation of Cookies Enriched with Osmodehydrated Wild Garlic from Nutritional and Sensory Aspects

**DOI:** 10.3390/foods13121941

**Published:** 2024-06-20

**Authors:** Vladimir Filipović, Milica Nićetin, Jelena Filipović, Alena Stupar, Jovana Kojić, Ivana Lončarević, Kosana Šobot, Jovanka Laličić-Petronijević

**Affiliations:** 1Faculty of Technology Novi Sad, University of Novi Sad, Bul. Cara Lazara 1, 21000 Novi Sad, Serbia; vladaf@uns.ac.rs (V.F.); ivana.radujko@tf.uns.ac.rs (I.L.); 2Institute of Food Technology in Novi Sad, University of Novi Sad, Bul. Cara Lazara 1, 21000 Novi Sad, Serbia; jelena.filipovic@fins.uns.ac.rs (J.F.); alena.tomsik@fins.uns.ac.rs (A.S.); jovana.kojic@fins.uns.ac.rs (J.K.); 3Faculty of Agriculture, University of Belgrade, Nemanjina 6, Zemun, 11080 Belgrade, Serbia; sobotkosana10@gmail.com (K.Š.); jovankal@agrif.bg.ac.rs (J.L.-P.)

**Keywords:** spelt cookies, osmodehydrated wild garlic, molasses, nutritive characteristic, sensory characteristics

## Abstract

In the present study, the nutritional and sensory properties of spelt cookies without wild garlic, cookies with fresh wild garlic, cookies with osmodehydrated wild garlic in sugar beet molasses, and cookies with osmodehydrated wild garlic in an aqueous solution of sucrose and salt were evaluated and compared. The tested cookie samples were characterized in terms of total antioxidative activity, the total content of phenols, flavonoids, and thiosulfates, the presence of dominant phenols, the content of betaine and dietary fiber, antioxidant activity after in vitro digestion, and sensory attributes for appearance, taste, smell, and texture. The results proved that the addition of wild garlic leaves osmodehydrated in molasses provided the cookies with the best nutritional and bioactive properties: 1.75 times higher total phenols content, 2.4 times higher total flavonoids content, 1.52 times higher total thiosulfates content, and 1.56 times higher betaine content, and a total quality increase of 54% compared to the control cookies. The cookies enriched with osmodehydrated wild garlic in molasses were rated as pleasant and acceptable, but also more complex compared to other cookies. The production of this nutritionally and sensory-improved cookie would contribute to expanding the assortment of flour confectionery products, especially for consumers who care about health and nutrition.

## 1. Introduction

Attractive taste, appearance, and texture, long shelf life, and affordable price are the attributes that make cookies a very popular food, enjoyed worldwide, among the population of all generations [1,2]. However, this type of product is often poor in minerals, bioactive compounds and dietary fiber [3]. Given the growing consumer awareness and demand for a healthier diet, the enrichment of specialty bakery products with bioactive components that have proven positive effects on human health is gaining importance [4]. Cookies represent a good matrix for the production of tasty and healthy food, because its nutritional value and functionality can be easily improved by adding high-value ingredients and changing the basic formula [2,5]. Several studies have demonstrated the enhancement of cookies’ nutritional value and functionality by incorporating antioxidant-rich raw materials, as well as fruits like peaches [6] and vegetables such as spinach [7]. Strategies for reducing the share of wheat flour in the composition of cookies, or its complete replacement with other types of flour, are also present. For this purpose, buckwheat, barley, spelt, rye, oat, flax, and amaranth flour are most often used [6,8,9,10].

Osmotic dehydration, in addition to being an effective method for reducing the water content in food raw materials (over 50%), also evinces advantages in terms of the economic, ecological, and nutritional aspects, such as the requirement of low process temperatures, low energy consumption, and minimal treatment of the initial raw materials [11,12]. Sugar beet molasses, a highly concentrated liquid mixture of sugar and numerous valuable compounds, such as minerals, proteins, betaine, and polyphenols, can be easily implemented as an osmotic dehydration agent [13,14]. Due to its availability and favorable nutritional composition, which enriches the dehydrated material, it has proven to be an excellent alternative to standard osmotic solutions such as aqueous solutions of sugar, salt, and their combination [14,15].

The following phenols contribute the most to the antioxidant capacity of sugar beet molasses: gallic acid [16,17], ferulic, vanillic, syringic and p-hydroxybenzoic acid [16,18,19], catechin [20], and kaempferol [19]. According to some authors, betaine, present in molasses with a share of up to 6%, is also a carrier of antioxidant capacity, protecting liver function, cells from apoptosis, and the body from the effects of free radicals [21].

The biggest challenge facing the food industry in the production of enriched cookies is that, by the addition of functional ingredients, the sensory quality is mostly affected, which impacts consumer acceptability [2,9]. In addition to maintaining an acceptable texture, color, and other sensory properties of cookies, it is also in line with consumer expectations that minimally processed foods are used for the production of cookies, without the use of synthetic antioxidants [1]. Recently, there has been a particular focus on the possibility that secondary products of the food industry and agro-industry residues, as a source of natural antioxidants and other potentially functional ingredients, can enrich different food products [22,23]. Furthermore, it is important to focus on the economy of the production process itself, which is reflected in lower energy costs and the use of alternative raw materials that are cheaper than commercially available ones [24]. 

Wild garlic (*Allium ursinum* L.) has long been known as a medicinal plant with exceptional antimicrobial and antioxidant properties [25]. Sulfuric and phenolic compounds are present in the leaves of wild garlic to a large extent, and the presence of numerous minerals, dietary fiber, volatile compounds, pigments, etc. contributes to their significant bioactive and nutritional potential [26,27]. Thiosulfates, among which allicin is the main representative, are responsible both for the characteristic taste and smell of wild garlic, and its antioxidant and therapeutic properties [27]. As an addition to cookies, primarily in terms of nutrition, wild garlic leaves are a promising raw material, with the limiting factors being the high water content and short seasonal availability of this plant [28]. By osmotic dehydration of wild garlic leaves, a dehydrated semi-product of prolonged durability is obtained, with a significantly higher content of dry matter, so it can be added in larger quantities to the cookie formula and be available for more extended periods for cookie production [29].

In previous research [29], a new, nutritionally enriched cookie based on spelt flour was created, with the addition of osmodehydrated wild garlic leaves whereby, at the same time, the secondary product of the sugar industry—sugar beet molasses—was revalued. Optimization of the cookies’ formula in terms of the selected quality parameters was performed. It was shown that the cookie with the optimal formula had improved texture, color, chemical, and mineral composition. Results obtained from previous research [29] provided a starting point for the current study, where the nutritional profile (antioxidant activity, content of total and individual phenols, total flavonoids, total thiosulfates, betaine content, content of dietary fibers, digestibility) and sensory profile (appearance, taste, odor, texture) of cookies with the addition of an optimal amount of wild garlic leaves osmodehydrated in sugar beet molasses was evaluated. The evaluation of the newly proposed cookies’ quality characteristics was carried out in comparison to the cookies with the addition of wild garlic leaves osmodehydrated in sodium chloride (NaCl) and sucrose solution, as well as cookies with and without the addition of fresh wild garlic leaves.

## 2. Materials and Methods

### 2.1. Material

Wild garlic, the plant raw material required for this research, was collected in April of 2023, in a forest area on the slopes of the Fruška Gora Mountain (45°08′34.6″ N; 19°36′55.0″ E). The undamaged leaves were handpicked, taken to the laboratory shortly after harvesting, washed, and used fresh in the process of osmotic dehydration.

For the preparation of the ternary solution, the following raw materials were used: sodium chloride (“So produkt”, Stara Pazova) and sucrose (“Crvenka”, Serbia), i.e., commercially available products (table salt and sugar), purchased in a local market, and distilled water. Sugar beet molasses was used as the second osmotic solution, with a dry matter content of 85.04%. Molasses was obtained from the sugar factory (“Crvenka”, Serbia) and stored in closed plastic containers at room temperature, before being used in the osmotic dehydration process.

The following materials were used for making cookies with or without the addition of fresh or osmotically dehydrated wild garlic leaves: whole spelt flour (Jevtić, Bačko Gradište); margarine (“Diamant”, Zrenjanin); sugar (“Crvenka”, Serbia); table salt (“So produkt”, Stara Pazova); baking soda (NaHCO_3_) as the leavening agent (“Aleva”, Novi Kneževac); oregano (“Aleva”, Novi Kneževac); tap water.

### 2.2. Osmotic Dehydration Procedure

An aqueous solution of NaCl and sugar (ternary solution) was prepared by completely dissolving 350 g of salt in 1 L of distilled water with the help of an electric stirrer (Talboys Instrument Corp. Emerson, Thorofare, NJ, USA). After that, sugar, in the amount of 1200 g, was added to the solution, and also completely dissolved in the second mixing phase [30]. Sugar beet molasses was used in the process without prior preparation. The wild garlic leaves were thoroughly washed and dried with paper towels, then cut with scissors into pieces approximately 1 × 1 cm in size and immersed in laboratory containers filled with osmotic solutions. Wild garlic leaves were dehydrated in an osmotic solution ratio of 1:20 (*w*/*w*) in order to prevent excessive dilution of the osmotic solutions during the dehydration process. A large material/solution ratio was also chosen so that the wild garlic leaves, as a bulky material, could be well submerged and covered with the solution. The osmotic dehydration process was carried out at room temperature and atmospheric pressure for four hours. Manual mixing was performed every 15 min, in order to better homogenize the solution and diffuse water from the submerged samples, and to improve the mass transfer that takes place during the process. After 4 h, the samples were separated from the osmotic solutions and washed with tap water to remove the solutions that remained on the surface of the leaves. Paper towels absorbed the residual water from the dehydrated samples’ surface. The samples were stored in the refrigerator until the cookies’ preparation.

### 2.3. Preparation of Cookie Samples

Cookies were prepared using the modified AACC 10-50D method [31], based on which the basic formula was taken. Formulas of control cookies (sample 1), cookies with the addition of fresh wild garlic leaves (sample 2), cookies with the addition of wild garlic leaves osmodehydrated in ternary solution (sample 3), and cookies with the addition of wild garlic leaves osmodehydrated in molasses (sample 4) are presented in Table 1. Cookie formulas for the experimental plan in this research were proposed based on the optimal cookie formula obtained in the previous research of Šobot et al. [29].

The maximum level of the addition of fresh wild garlic was 2.5% of dry matter (d.m.) calculated on the amount of dry matter of flour, because fresh wild garlic contains only 8.4% of dry matter; thus, this addition incorporates the prescribed amount of water in the dough. Since the osmotically dehydrated leaves of wild garlic contain a much higher percentage of dry matter (30% for osmodehydrated wild garlic in ternary solution and 57% for osmodehydrated wild garlic in molasses), it is possible to add them in larger doses, but due to the comparability of the results, they were added in amounts that are equal in terms of dry matter (2.5%), i.e., 4.84 g. In the formula of cookie samples in which osmodehydrated wild garlic was included, the amount of added water was reduced by the amount of water originating from osmodehydrated wild garlic, so the total amount of water presented in all test formulas was the same.

Preparing cookies with and without fresh or osmodehydrated wild garlic leaves proceeded by placing all the measured ingredients, except water, in the mixing bowl and mixing for 3 min, then adding the water and mixing the dough for 2 min. The resulting dough was rounded and placed in a refrigerator to rest at 8 °C for 30 min. After that, the dough was laminated to a uniform height of 7 mm and cut to a uniform size of ϕ = 60 mm. The formed dough was placed in an oven preheated to 200 °C and baked for 10 min. The baked cookies (Figure 1) were allowed to cool and rest in controlled ambient conditions (21 °C and 60% relative humidity) for 24 h before further analysis.

### 2.4. Determination of the Bioactive Compounds

#### 2.4.1. Preparation of Extracts

Before analysis, the cookie samples were dried in a lyophilizer (Christ ALPHA 1-2 LDPLUS, Osterode am Harz, Germany), chopped, homogenized, and stored in a refrigerator at 4 °C until use in the appropriate analysis. The extracts were prepared using 80% methanol, in a ratio of 1:10 (m/V), followed by extraction at room temperature with shaking for 24 h. Subsequently, ultrasonic extraction was performed using an ultrasonic bath (EUP540A, Euinstruments, Paris, France) for 10 min. Then, the samples were filtered and used for further analyses.

#### 2.4.2. Spectrophotometric Determination of the Content of Total Polyphenols

In the obtained extracts, the content of total phenols (TPC) was determined by the known spectrophotometric method according to Folin–Ciocalteu [32]. TPC was calculated based on the calibration curve of the standard gallic acid solution and is expressed as gallic acid equivalent (GAE) in 100 g of dry matter of the tested sample. Analysis was performed in triplicate.

#### 2.4.3. Spectrophotometric Determination of the Content of Total Flavonoids

Determination of the content of total flavonoids (TF) was carried out using the spectrophotometric method according to Markham [32]. TF was calculated based on the calibration curve of the standard catechin solution and is expressed as catechin equivalent (CE) in 100 g of dry matter of the tested sample.

#### 2.4.4. Spectrophotometric Determination of the Content of Total Thiosulfates

The determination of total thiosulfates was carried out based on the method described by Han et al. [33], with modifications. Freeze-dried samples (2.5 g) were extracted with 20 mL of HEPES (4-(2-hydroxyethyl)-1-piperazineethanesulfonic acid) buffer (50 mM, pH 7.5) with stirring on an orbital mixer (300 rpm min), for 15 min, at room temperature. After extraction, the obtained extract was centrifuged at 10,000 rpm for 10 min and filtered. The resulting extract (1 mL) was mixed with 1 mL of L-cysteine (5 mM in HEPES buffer). The formed reaction mixture was diluted to 50 mL and left for 15 min at room temperature. Then, 9 mL of the obtained mixture was reacted with 1 mL of DTNB solution (1.5 mM in HEPES buffer). At a wavelength of 412 nm, the absorbance was read after 15 min.

For the blank test, a pure solvent (HEPES buffer) was used instead of the extract. The content of total thiosulfinates is expressed as the equivalent of allicin (EA) in 100 g of dry matter of the tested sample. Analysis was performed in triplicate

#### 2.4.5. Identification and Quantification of Polyphenols by Liquid Chromatography (HPLC-DAD)

The qualitative and quantitative profile of phenolic compounds was determined according to the method presented in the study of Mišan et al. [34]. This method involves high-pressure liquid chromatography, which takes place on an apparatus (Agilent 1200 series, Paolo Alto, CA, USA) using an Agilent, Eclipse XDB-C18 column (1.8 μm, inner diameter 4.6 × 50 mm; size loading 1.8 μm) and diode array detector (Agilent, USA). Analysis was performed in duplicate.

### 2.5. Determination of the Antioxidative Activity

The antioxidative activity of all tested cookie samples was determined using spectrophotometric methods of ABTS and DPPH radical neutralization. The preparation of the tested extracts and the procedure were conducted according to Popović et al. [35].

The obtained values of the antiradical activity of the tested samples are also expressed as sample concentrations (mg/mL) required for inhibition of 50% of the initial concentration of ABTS and DPPH radicals. Analysis was performed in triplicate.

### 2.6. Determination of Betaine Content

Betaine quantification in the samples was performed using the high-performance liquid chromatography method [36]. Analysis was performed in duplicate.

### 2.7. Determination of Dietary Fibre Content

The proportion of total and insoluble dietary fiber in the cookie samples was determined according to the standard method AOAC 991.43 [37]. Analysis was performed in duplicate.

### 2.8. In Vitro Digestion

The method of in vitro digestion of cookie samples was performed based on the modified method described by Minekus et al. [38]. This method is based on three phases that simulate mouth, stomach, and intestine conditions. The preparation of simulated juices of the gastrointestinal tract took place according to the following recipe: simulated saliva juices (KCl-15.1 mM, KH_2_PO_4_-3.7 mM, NaHCO_3_-13.6 mM, MgCl_2_(H_2_O)6-0.15 mM, (NH_4_) 2CO_3_-0.06 mM), stomach (KCl-6.9 mM, KH_2_PO_4_-0.9 mM, NaHCO_3_-25 mM, NaCl-47.2 mM, MgCl_2_(H_2_O)6-0.1 mM, (NH_4_) 2CO_3_-0.5 mM), and intestines (KCl-6.8 mM, KH_2_PO_4_-0.8 mM, NaHCO_3_-85 mM, NaCl-38.4 mM, MgCl_2_(H_2_O)6-0.33 mM). The pH of the juices was adjusted with 0.1M NaOH and 6M HCl to pH 7 for salivary and intestinal juices, and pH 3 for gastric juices.

According to Čakarević [39], in the obtained samples, the antioxidative activity was determined using the spectrophotometric method of ABTS radical neutralization.

### 2.9. Descriptive Sensory Analysis

The sensory analyses were conducted by the 8-member panel comprising employees of the Faculty of Technology and Institute for Food Technology, who had previous experience testing different food products. A panel of these trained assessors (five women and three men) met the criteria of the ISO standard (ISO 8586:2012) [40], from the sensory and technical analysis department of the accredited laboratory of the Institute of Food Technology, Novi Sad, Serbia, and was formed according to the appropriate standard, ISO 6658:2017 [41], to conduct a descriptive sensory evaluation of the cookie samples. The leading evaluator previously selected descriptors for the sensory profiling of the cookie samples and performed further adjustment with the rest of the panelists to better define the sensory profile of the cookies. The final list consisted of four descriptors, namely a descriptor that characterizes the appearance of the cookies (color intensity and surface appearance), a descriptor that characterizes deviation from the standard taste, a descriptor that shows a deviation from the standard odor, and a descriptor that defines textural properties (sensory hardness and brittleness, fracturability, and mastication). A nine-point scale measured the intensity of each selected descriptor, where 1 describes the lowest intensity and 9 the highest intensity.

The sensory evaluation of the cookie samples was performed 24 h after baking in the sensory analysis laboratory of the Institute of Food Technology, Novi Sad, Serbia, designed according to the ISO 8589:2007 standard [42]. Cookie samples were placed in front of the evaluators on white plastic plates in random order, coded with random three-digit codes (created by free online randomizer software), at room temperature (20 ± 2 °C), and each panelist evaluated all four samples. The method of eating consisted of thoroughly chewing and swallowing the whole cookie sample. After testing each cookie sample, and prior to testing the successive sample, the evaluators rinsed their mouths with water, as a an oral cleansing step. Each panelist carried out two tastings. The sensory analysis was conducted in strict compliance with the ethical principles stated in the Declaration of Helsinki. All members of the evaluation team possessed the necessary qualifications and expertise in the relevant field. Information for the descriptive sensory analysis assessors and the cookie sample descriptive sensory analysis form are presented in Appendix A.

### 2.10. Methods of Statistical Analysis

#### 2.10.1. Correlation Analysis

The color correlation diagram for nutritive and descriptive sensory characteristics’ mean values (31 in total) of cookies with and without fresh and osmodehydrated wild garlic addition was designed and drawn using R software v.4.3.0 (64-bit version). The following settings were used for the correlation presentation between tested samples’ responses: the corrplot instruction, “circle” method, upper type.

#### 2.10.2. Principle Component Analysis

Principal component analysis (PCA) was used for pattern recognition in the analyzed data. PCA calculation was performed using Microsoft Excel ver. 2016. (Microsoft Corporation, Redmond, WA, USA), with the XLSTAT Version 2014 5.03 Add-in (Lumivero, Denver, CO, USA).

#### 2.10.3. Analysis of Variance

Analysis of variance (ANOVA) was used to determine the significance of variations in all tested quality responses of the examined cookies’ samples. STATISTICA 12.0 software (2013) (StatSoft Europe, Hamburg, Germany) was used for ANOVA analysis.

#### 2.10.4. Z-Score Analysis

The Z-score analysis uses min-max normalization for different cookie samples’ response values, where response values are mathematically transformed from their original unit system to a new dimensionless unit system, for comparisons and further mathematical calculations of these different responses [6].

Detailed calculations of the Z-score values are presented in Appendix A.

Z-score calculations were performed using Microsoft Excel ver. 2016. (Microsoft Corporation, Redmond, WA, USA).

## 3. Results and Discussion

### 3.1. Nutritional Quality Characteristics of Cookies

#### 3.1.1. Cookies’ Total Antioxidant Activity and the Content of Total Phenols, Flavonoids, and Thiosulfates

The change in total bioactive compounds (phenols, flavonoids, and thiosulfates) and antioxidant activity in cookie samples enriched with fresh and osmodehydrated wild garlic leaves in all of the used osmotic solutions (samples 2, 3, and 4) was monitored and compared to that of the control cookie (sample 1). The obtained results are shown in Table 2.

Compared to the cookies of the basic formula, in which the content of total phenols was 0.24 g GAE/100 g d.m., the addition of fresh wild garlic statistically significantly increased the content of total phenols to 0.30 g EGK/100 g d.m., i.e., by 25%. With the addition of wild garlic leaves osmodehydrated in ternary solution, the increase in the content of total phenols was less pronounced, amounting to a statistically insignificant 5%. However, the addition of osmodehydrated wild garlic leaves in molasses contributed to a statistically significant increase in the quantity of total phenols in the cookie to 0.42 g EGK/100 g d.m., i.e., the content of total phenols increased by 75% compared to that of the control samples. This is in accordance with research conducted by Cvetković et al. [43], who stated that molasses contributed to an increase in the total phenolic content, while the ternary solution reduced the phenolic content in cabbage.

The results presented in Table 2 further demonstrate that the cookie enriched with osmodehydrated wild garlic in molasses exhibited a statistically significant increase in total flavonoids content, of 2.4 times, compared to the control cookie. The addition of fresh wild garlic leaves to the basic formulation of the control sample increased the value of total flavonoids by a statistically significant 1.7 times and the addition of osmodehydrated wild garlic leaves in ternary solution by a statistically insignificant 1.3 times. The best result regarding the content of total flavonoids obtained for the samples with osmodehydrated wild garlic in molasses as an ingredient indicates that part of the flavonoids from the molasses contributes to the improvement of the composition of the cookies, since Nićetin et al. [44] confirmed that sugar beet molasses contains 86.07 mg/100 g d.m. of catechin and 98.03 mg/100 g d.m. of kaempferol.

By incorporating wild garlic into the cookie recipe, cookies were enriched with thiosulfates, contributing a unique flavor profile and potential health benefits to the cookies. The highest statistically significant increase in total thiosulfates compared to the initial content was contributed by the addition of osmodehydrated wild garlic in molasses (1.52 times), followed by the addition of fresh wild garlic (1.44 times). The least influence was observed by the addition of osmodehydrated wild garlic in ternary solution (1.12 times).

The addition of fresh wild garlic leaves resulted in increased values of antioxidant activity obtained from both ABTS and DPPH methods (concentration required to inhibit 50% of radicals (IC_50_); lower ABTS and DPPH values indicating higher antioxidative activity). When comparing samples of cookies with the addition of osmodehydrated wild garlic leaves in ternary solution, to samples of cookies with fresh wild garlic, statistically insignificantly lower values of antioxidant activity were obtained. This is in agreement with previous results published by the authors Knežević et al. [30] and Nićetin et al. [45], which showed that the total antioxidant capacity of nettle and celery leaves was reduced by 10–20% after osmotic dehydration at room temperature in an aqueous solution of sucrose and sodium chloride. On the other hand, samples of cookies containing osmodehydrated wild garlic leaves in molasses, at the same levels of addition, showed statistically insignificantly higher values of antioxidant capacity determined by ABTS and DPPH methods, compared to samples of cookies with fresh wild garlic. The lowest concentration required to inhibit 50% of ABTS and DPPH radicals, 12.37 mg/mL and 62.80 mg/mL, was shown by sample 4, which contains 2.5% wild garlic leaves osmodehydrated in molasses. This result is consistent with the findings of other authors [13,14,15,16,17], who already confirmed that after the process of osmodehydration in sugar beet molasses, the total antioxidant capacity of various foods increased.

#### 3.1.2. Cookies’ Polyphenols Liquid Chromatography Identification and Quantification

The qualitative and quantitative profile of the most abundant detected phenolic compounds in the tested samples, extracted using methanol and the ultrasonic extraction technique and the application of the HPLC-DAD technique on the obtained extracts, is shown in Table 3. The chromatograms obtained from the analysis were carefully examined to identify the prominent peaks, and the phenolic compounds present in the samples were determined. However, it should be noted that certain peaks remained unidentified, likely due to the presence of phenols in bound form, such as glycosides or polymers, which pose challenges in their detection and quantification using the obtained HPLC chromatograms.

The results of the HPLC analysis show a clear difference in the chemical composition of the cookie samples with fresh and osmodehydrated wild garlic. In the control cookie without wild garlic, only catechin, which comes from spelt flour, was detected. The addition of fresh wild garlic caused a statistically significant increase in the initial concentration of catechin, as well as the detection of a certain concentration of kaempferol derivatives, chlorogenic and ferulic acid. This is in compliance with a report by authors who examined the phenolic profile of wild garlic and proved that kaempferol derivatives were the most dominant [28]. Carutenuto et al. [46] identified five kaempferol glycosides in the ethanolic extracts of wild garlic, among which two were bound by *p*-coumaric and ferulic acids. Wu et al. [47] were able to isolate seven kaempferol glycosides from the ethanolic extract of wild garlic, while Oszmiański et al. [48] identified twenty-one phenolic compounds in wild garlic leaves and all detected compounds were kaempferol derivatives. In accordance with the HPLC chromatograms of wild garlic extracts by Pavlović et al. [49], the highest amounts of 3,7-kaempferol-diglucoside and kaempferol 3-glucoside were found in the methanol extract, and ferulic acid was detected in traces.

According to the concentration of detected compounds in cookies with fresh wild garlic leaves (sample 2), kaempferol derivatives were followed by catechin, ferulic, and chlorogenic acids. In the research of Stupar et al. [28], the presence of catechins was confirmed in the extract of wild garlic, obtained by extraction with subcritical water. The presence of ferulic acid is in accordance with the findings of Vlase et al. [50], who detected ferulic and *p*-coumaric acids in the hydrolyzed and non-hydrolyzed ethanolic extract of wild garlic, while they detected kaempferol only in the hydrolyzed extract. By analyzing the phenolic profiles of several *Allium* species, Parvu et al. [51] identified ferulic and *p*-coumaric acid in all samples and concluded that *Allium ursinum* is the richest source of kaempferol derivatives compared to other species. Krivokapić et al. [52] identified and quantified kaempferol 3-O-glucoside as the most abundant phenolic component (376 mg/100 g of dry extract), followed by *p*-coumaric acid, ferulic acid, free kaempferol, and ursolic acid in the methanolic extract of wild garlic. Concerning the available scientific research dealing with the analysis of the phenolic profile of wild garlic, which is quite scarce, chlorogenic acid was identified for the first time by the HPLC analysis carried out within the presented research.

The obtained results for samples 3 and 4 (Table 3) show that the osmotic dehydration process affected the change in the initial concentration of catechin, kaempferol derivatives, and chlorogenic and ferulic acids. Osmotic dehydration in ternary solution affected the reduction in all detected phenolic compounds, while osmodehydration in molasses contributed to a statistically significant increase in their concentration, compared to the sample with fresh wild garlic. In relation to all four analyzed cookie samples, the cookie with the addition of osmotically dehydrated wild garlic in molasses had the highest value of catechin, kaempferol derivatives, and chlorogenic and ferulic acids. The contribution of molasses to the increase in the quantity of phenolic components present in wild garlic can be explained by the fact that molasses itself is rich in phenols. HPLC-DAD analysis by the authors Nićetin et al. [44] confirmed that sugar beet molasses contains 86.07 mg/100 g d.m. catechin, 98.03 mg/100 g d.m. kaempferol, 156.06 mg/100 g d.m. ferulic acid, and 36.82 mg/100 g d.m. chlorogenic acid. In the study of Cvetković et al. [43], after osmotic dehydration of cabbage in molasses, the content of kaempferol, catechin, and *p*-coumaric, caffeic, and gallic acids increased. Filipčev et al. [17,53] confirmed that the addition of sugar beet molasses in gluten-free cookies contributed to the increase in most of the identified phenolic compounds, among which are catechin, *p*-coumaric acid, ferulic acid, and gallic acid.

#### 3.1.3. Cookies’ Betaine, and Total and Insoluble Dietary Fiber Contents

In the literature of the previous research, there are data showing that bakery and flour-confectionery products, especially those based on wheat flour, contain low levels of betaine [54]. With the aim of increasing the level of betaine in these products and consequently improving the daily intake of betaine in the diet, it is suggested that existing product formulas should be modified by including betaine-rich ingredients such as sugar beet molasses, amaranth, spelt, quinoa, and millet in their composition [36].

In the control cookies without wild garlic, a significant amount of betaine of 538.79 mg/100 g was detected, which is probably fully contributed by whole spelt flour. In the research of Kojić et al. [36], spelt flour was evaluated as the richest in betaine, compared to the flours of other cereals. Fresh wild garlic added to the cookie formula statistically significantly reduced the total betaine content in the control cookie by about 9%. Osmodehydrated wild garlic in ternary solution reduced the total betaine content in the cookies, at the same level of statistical significance, as the fresh wild garlic addition. On the other hand, the addition of wild garlic osmodehydrated in molasses contributed to a statistically significant increase in betaine content by 56%, compared to the control cookies.

A higher addition of osmodehydrated wild garlic in molasses, which is possible up to 30%, would increase the betaine content by about 18 times. The obtained results are consistent with previous research, which showed that the enrichment of cereal-based products with sugar beet molasses, even at lower doses, significantly increases the betaine content and can be considered a practical way to improve the health impact of the product [53]. The addition of sugar beet molasses to the usual cookies’ formula increased the betaine content from 11 mg/100 g d.m. to 473 mg/100 g [17]. In the gluten-free cookies enriched with 30% molasses (calculated on flour), the level of betaine increased approximately 64 times compared to the control cookies [54].

A portion of 100 g cookies made from whole grain spelt flour with the addition of wild garlic osmodehydrated in molasses provides an intake of 843 mg of betaine. This amount exceeds the recommended daily portion of 500 mg and is a little more than half of the recommended daily amount of betaine (1500 mg/day). Consuming a portion of 100 g cookies made from whole grain spelt flour with the addition of wild garlic osmodehydrated in molasses might contribute to normal homocysteine metabolism, according to European regulations (Commission Regulation EU No. 432/2012) [55].

The results presented in Table 4 indicate that adding wild garlic leaves (fresh and osmodehydrated in both solutions) affected the statistically significant change in the content of total and insoluble dietary fibers in the obtained cookie samples. It is observed that all four types of cookies contained a significant percentage of dietary fiber, with the cookie enriched with fresh wild garlic being the most abundant, having a total fiber content of 17.03% and insoluble fiber of 6.96%. This result indicates that wild garlic leaves are a good source of soluble and insoluble fiber, and they can significantly enrich cookies with these carbohydrates important from a nutritional and health perspective. It is known that dietary fiber has a potential role in reducing the risk of hypertension, obesity, diabetes, heart disease, stroke, and certain gastrointestinal disorders, which is why the recommended dose is in the interval of 25–35 g of fiber per day [55].

The addition of fresh wild garlic leaves to the basic formula of cookies (2.5% d.m. based on flour) statistically significantly increased the content of total dietary fiber by 24.67% and insoluble fiber by 15.6% compared to the control samples. The same amount of wild garlic osmotically dehydrated in ternary solution statistically significantly reduced the initial content of total dietary fiber by 12.17% and that of cellulose by 10.6%. The decrease in fiber content compared to the control cookie is due to the fact that part of the flour is replaced with osmotically dehydrated wild garlic leaves, which have a higher proportion of sugar and salt in their composition of dry matter than wild garlic’s components. Conversely, the addition of osmodehydrated wild garlic in molasses contributed to a statistically significant increase in the amount of total dietary fiber, of 10.27%, while the content of insoluble fiber remained almost unchanged. This can be explained by the fact that, with the addition of osmodehydrated wild garlic leaves in molasses to the cookie formula, a smaller part (about 30%) of soluble and insoluble fibers from wild garlic was introduced, and a larger part of dry matter from molasses (about 70%), which contains a certain percentage of soluble dietary fiber, but not insoluble fiber [17].

Although the addition of fresh wild garlic in the amount of 2.5% d.m. directly contributed to obtaining a cookie with the highest percentage of total dietary fiber, by 1.24 times more than in the control cookies, it should be taken into account that this is the maximum possible addition of fresh wild garlic without damaging the texture of the dough. In the case of adding osmodehydrated wild garlic leaves to the cookie formula, higher doses are possible, so an improved effect on the fiber composition of the cookies is expected. The maximum possible addition of osmodehydrated wild garlic leaves in molasses (30% d.m. based on flour, i.e., 101.8 g) would increase the amount of total dietary fiber by 12.3 times compared to control cookies.

#### 3.1.4. Cookies’ In Vitro Digestion

The application of in vitro digestion as a model gives insight into the behavior of bioactive compounds from food after passing through the human digestive tract. During digestion, the active compounds of the sample are exposed to various environmental conditions, such as pH changes, the presence of gastric and intestinal juices, and the action of enzymes, so it is necessary to examine their influence on the bioactivity of the sample. In addition to digestibility, it is important that food retains or releases some biological activity that will not be impaired after digestion [56].

As part of this research, the digestion process was simulated in order to evaluate the bioactive properties and resistance of the active compounds, upon ingestion. The bioactive properties of all cookie samples, which passed through the conditions of the gastrointestinal tract, were tested. The antioxidant activity of these four samples before and after digestion is shown in Figure 2.

The presented results of measuring antioxidant activity (AA) prove that all samples after digestion possessed certain biological activities.

A similar curve trend is observed for all samples, where the antioxidant activity increases over time and reaches a maximum value after 10 min. Based on the presented results, it can be seen that the lowest ABTS radical removal activity after digestion was recorded in the control cookies (40%). Samples enriched with fresh and osmotically dehydrated wild garlic had higher antioxidant activity compared to the control sample. The highest activity was measured for the cookies enriched with osmodehydrated wild garlic in molasses (about 63%), and for the cookie with osmodehydrated wild garlic in ternary solution, it was slightly lower (about 56%). The hydrolysate of cookies with fresh wild garlic also had a pronounced ability to neutralize ABTS+ radicals, where AA reached a value of 60%.

The increase in the antioxidant activity of the hydrolysate of cookie samples during passage through the digestive tract may be a consequence of the release of the content of phenolic components in enzymatic processes. In the research of Čakarević [39], it was shown that all tested cookies enriched with encapsulate (protein isolate and beetroot juice) possessed antioxidant activity after digestion, where the highest activity (about 80%) was measured at the highest addition of encapsulate (30%). It was assumed that the increase in activity occurs due to the presence of phenolic components from beetroot juice and peptides that are released from proteins.

These results indicate that the tested cookie samples after consumption, i.e., exposure to the influence of the gastrointestinal tract, show a good free radical scavenger potential. Therefore, enriching cookies with wild garlic, and foremost wild garlic osmodehydrated in molasses, can increase the antioxidant activity of the product during its consumption.

### 3.2. Descriptive Sensory Characteristics of Cookies

The results of descriptive sensory analysis of the evaluated cookie samples are presented in Table 5. It can be seen that the addition of wild garlic leaves, both fresh and osmodehydrated, statistically significantly affected the descriptors of appearance, taste, odor and texture, compared to the cookies without the addition of wild garlic.

Cookies with added fresh and osmodehydrated wild garlic deviated from the characteristic appearance of cookies without these ingredients, from the initial score of 8 to 7. Additions of fresh and osmodehydrated wild garlic in both osmotic solutions contributed to a statistically significant increase in the color intensity of the obtained cookie samples, but correspondingly to a statistically significant decrease in color uniformity compared to the control cookie samples. With the addition of wild garlic leaves (samples 2, 3, and 4) to the cookie formula, the descriptor for the appearance of the surface changed statistically significantly, from smoother, for cookies without these components, to rougher.

Compared to the control cookies, the addition of osmodehydrated wild garlic in ternary solution statistically significantly intensified the descriptor for bitter taste (1–7.5), while fresh wild garlic statistically significantly enhanced the herbaceous taste descriptor (1–9). The osmodehydrated wild garlic in molasses was the solely enriching component that had a statistically significant increasing effect on the descriptors for caramel taste (2–5). The spiciness of the taste was equally contributed by the addition of fresh and osmodehydrated wild garlic in ternary solution (1–8), and slightly less by osmodehydrated wild garlic in molasses (1–7.5). These results indicate that the cookies with osmodehydrated wild garlic in molasses can be pleasant and acceptable to consumers, but also more complex in terms of taste compared to the taste of the other analyzed cookie samples. Mišan et al. [57] reported that the added quantity of herbal mixtures (mint, parsley, buckthorn, caraway) does not adversely affect the sensory quality of the cookies but provides a specific aroma accepted by sensory panelists. Additionally, other authors [58,59,60,61,62] confirmed that herbal cookies had acceptable sensory attributes.

A statistically significant increase in the caramel odor was found to be characteristic only for cookie samples with osmodehydrated wild garlic leaves in molasses, which is a consequence of a certain content of molasses in the final product. Sample 3 was rated as the one with the most pronounced pungent odor. The spicy odor was more pronounced in the cookie samples with fresh wild garlic, compared to those with the addition of osmodehydrated wild garlic leaves, while the addition of osmodehydrated wild garlic in molasses contributed the most to the herbal odor.

In terms of texture, it can be seen that samples with osmodehydrated wild garlic in molasses showed the highest hardness, while the sample with fresh wild garlic had higher fracturability and mastication in comparison to the samples with osmodehydrated wild garlic. Cookies with the addition of osmodehydrated wild garlic leaves in molasses had a statistically significantly higher sensory firmness, fracturability, and mastication, and a statistically significantly lower brittleness compared to the control cookies.

### 3.3. Statistical Analysis

#### 3.3.1. Correlation Analysis

Correlation analysis of nutritive and descriptive sensory responses was conducted to indicate interactions and correlations of all tested responses of cookie samples of different formulations. The obtained results of the correlation analysis are valid only for the tested sets of samples, and do not indicate general correlations between responses. Figure 3 presents the color correlation diagram between 31 responses of the nutritive and descriptive sensory analysis of the tested cookies. Correlation coefficients’ values between any two tested responses can be easily identified due to the determination by color, color intensity, and the size of the circles (blue for positive and red for negative correlation, the intensity of color, and the bigger circle size for the proximity to 1 or −1).

Correlation analysis results present a high level of positive correlation between all the responses of total phenolics, flavonoids, and thiosulfates, individual dominant polyphenols, digestibility, and some of the descriptive sensory responses (color intensity, surface appearance, bitter, spicy, caramel and herb odor and smell, and fracturability and mastication), indicating that addition of wild garlic, especially that osmodehydrated in molasses, affected the polyphenol profile and the stated descriptive sensory responses with a high level of correlation. On the correlation diagram, responses of antioxidative activity (ABTS and DPPH), expressed as IC_50_ values (a lower value indicates a higher antioxidative value), display a negative correlation with the responses of total phenolics, flavonoids, and thiosulfates, and individual dominant polyphenols, hence indicating a high correlation between antioxidative activity and all groups of polyphenol compounds of the tested cookies (with the addition of either fresh or osmodehydrated wild garlic).

Other cookies’ nutritive responses (betaine, total dietary fiber, and insoluble dietary fiber) were poorly correlated with all other responses, except for betaine with total content of phenolics, and total dietary fiber and insoluble dietary fiber with total content of thiosulfates.

The descriptive sensory response of standard appearance, which describes control cookies (without wild garlic addition) with high values, was negatively correlated with the most descriptive sensory responses (color intensity, surface appearance, all taste and odor responses, fracturability, and mastication), underlying the difference between the sensory responses of cookies with and without wild garlic addition.

The results of the correlation diagram confirm the previously discussed individual responses, where the addition of osmodehydrated wild garlic (especially by molasses) to the formula of the cookies affected most of the nutritive responses, correlating them with significant changes in cookies’ descriptive sensory responses.

#### 3.3.2. Results of the PCA

PCA was performed in an effort to analyze the correlation structure [63] between 31 experimentally determined responses and four different cookies samples. For the purpose of the visualization of the data trends and the discriminating efficiency of the used descriptors, a sample scatter plot was produced, plotting the first two principal components of the data matrix, with the first principal component as the x-axis and the second as the y-axis, Figure 4.

A significant separation of four tested cookie samples is achieved due to the different quality responses, which can be visually determined from the presented scatter plot. The cookie samples’ position on the scatter plot was primarily determined by the wild garlic addition (control cookies, sample 1, without wild garlic addition was located in the area of negative first principal component values, while the addition of the wild garlic moved the cookie samples’ position to the areas of positive first principal component values). The dehydration method type determined the cookie samples’ position along the second principal component, where cookies’ samples with fresh wild garlic addition were located in the area of negative second principal component values and cookie samples with the addition of osmodehydrated wild garlic in molasses were located in positive second principal component values areas.

High values of standard appearance, ABTS, DPPH (low antioxidative activity), color uniformity, and brittleness, and low values of all other tested responses, characterized the control cookie sample. Cookie samples with fresh wild garlic addition were mostly characterized by high values of total and insoluble dietary fibers and fracturability, where cookie samples with the addition of osmodehydrated wild garlic in ternary solution were mostly characterized by having high values of bitter, spicy, and herb taste, spicy odor, and color intensity. The difference between these two samples was not so profound, since a similar intensity of the same vectors (responses) characterized both samples, signifying the effect of the wild garlic addition to these cookies’ quality attributes. Cookie samples with osmodehydrated wild garlic in molasses were characterized by high values of total phenolics, flavonoids, and thiosulfates, individual dominant polyphenols, betaine, digestibility, caramel taste and odor, indicating the effect of molasses on these cookies’ quality characteristics.

Quality testing indicated that the first two principal components accounted for 89.72% of the total variance, and hence can be considered sufficient enough for the given data description. The most dominant contributors of the tested responses to the first principal component were individual dominant polyphenols, ABTS, DPPH, digestibility, most of the descriptive sensory responses, and total phenolics, flavonoids, and thiosulfates. However, the contribution of the most significant responses to the second principal component, in decreasing order, was betaine content, hardness, caramel taste and odor, insoluble dietary fiber, total phenolics, and fracturability.

#### 3.3.3. Results of the Z-Score Analysis

Z-score analysis was performed to quantify the differential total quality characteristics of the four tested cookie samples, based on the 31 investigated nutritive and descriptive sensory characteristics. Figure 5 presents the results of the Z-score analysis of four cookie samples, with and without the addition of fresh and dehydrated wild garlic.

The presented results show Z-score segments S_1_ to S_7_, corresponding to Z-score results for antioxidative activity, individual phenolic compounds, dietary fibers and digestibility, descriptive sensory analysis of appearance, descriptive sensory analysis of taste, descriptive sensory analysis of odor, and descriptive sensory analysis of texture, respectively.

The presented results indicate that the addition of wild garlic (either fresh or osmodehydrated) significantly increased all segment Z-score values, except for appearance. Incorporation of fresh wild garlic in comparison to the wild garlic dehydrated in ternary solution into the cookies’ formula provided better segment score results for antioxidative activity, individual phenolic compounds, dietary fibers, digestibility and taste, while segment scores values for odor and texture were in close proximity. Cookie sample 4, with the addition of osmodehydrated wild garlic in molasses, was characterized by the highest segment score values, which were significantly higher than in other samples, for antioxidative activity, individual phenolic compounds, taste, and odor. The only segment score where sample 4 was inferior in comparison to the other samples was descriptive sensory analysis of texture (value of 0.53 for sample 4, 0.63 and 0.64 for samples 2 and 3, respectively).

Total Z-score values were set to mathematically combine all segment Z-scores with the following contributions: 50% of nutritive response values (20% for S_1_ and S_3_, 10% for S_2_) and 50% for descriptive sensory characteristics (10% for S_4_, S_6_, and S_7_, 20% for S_5_). In that manner, total Z-score values point at the optimal combination of all tested cookies’ nutritive and descriptive sensory responses. The results of the total Z-score indicate that the addition of osmodehydrated wild garlic in molasses to the cookies’ formula significantly increased total nutritive and sensory quality, obtaining a value of 0.75 out of maximal value 1.

## 4. Conclusions

Based on the presented results, it can be concluded that cookies enriched with wild garlic leaves osmodehydrated in molasses proved to be superior in terms of most of the tested nutritional and sensory characteristics. These cookies had 1.75 times higher total phenols content, 2.4 times higher total flavonoids content, 1.52 times higher total thiosulfates content, and 1.56 times higher betaine content compared to the control cookies. The results confirmed that the incorporation of wild garlic osmodehydrated in molasses in cookies contributed to the increase in all phenolic compounds identified in cookies with fresh wild garlic, among which were catechin, three kaempferol derivatives, chlorogenic acid, and ferulic acid. Cookie samples’ in vitro testing showed a good free radical scavenging potential, especially in the samples with wild garlic osmodehydrated in molasses.

The results of the correlation analysis highlighted the positive effect of the addition of osmodehydrated wild garlic in molasses to the cookies formula, mostly on the nutritive responses, concurrently manifesting significant changes in cookies’ descriptive sensory responses. PCA results confirmed a positive effect of the addition of wild garlic to the cookies’ nutritive characteristics and descriptive sensory response alterations. The results of the Z-score analysis provided insight into total cookies’ quality increase by the addition of osmodehydrated wild garlic in molasses, from a 12 percentile point increase in comparison to the cookies with the addition of fresh wild garlic, to a 54 percentile point increase in comparison to the control cookies.

Future research is required to define the optimal level of addition of wild garlic leaves osmodehydrated in molasses to obtain the cookie formula with the most favorable nutritional profile, without compromising the technological and sensory quality, in order to obtain a finished product ready for the market.

## Figures and Tables

**Figure 1 foods-13-01941-f001:**
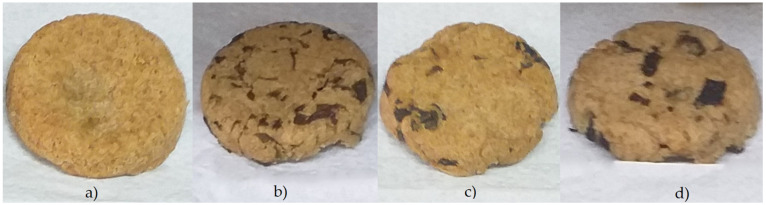
Control cookies (**a**), cookies with the addition of fresh wild garlic leaves (**b**), cookies with the addition of wild garlic leaves osmodehydrated in ternary solution (**c**), cookies with the addition of wild garlic leaves osmodehydrated in molasses (**d**).

**Figure 2 foods-13-01941-f002:**
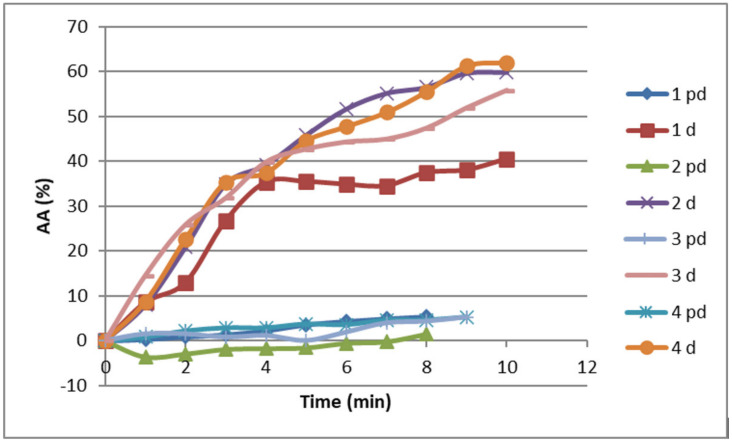
Antioxidant activity of cookie samples (1–4) prior (pd) and after digestion (d).

**Figure 3 foods-13-01941-f003:**
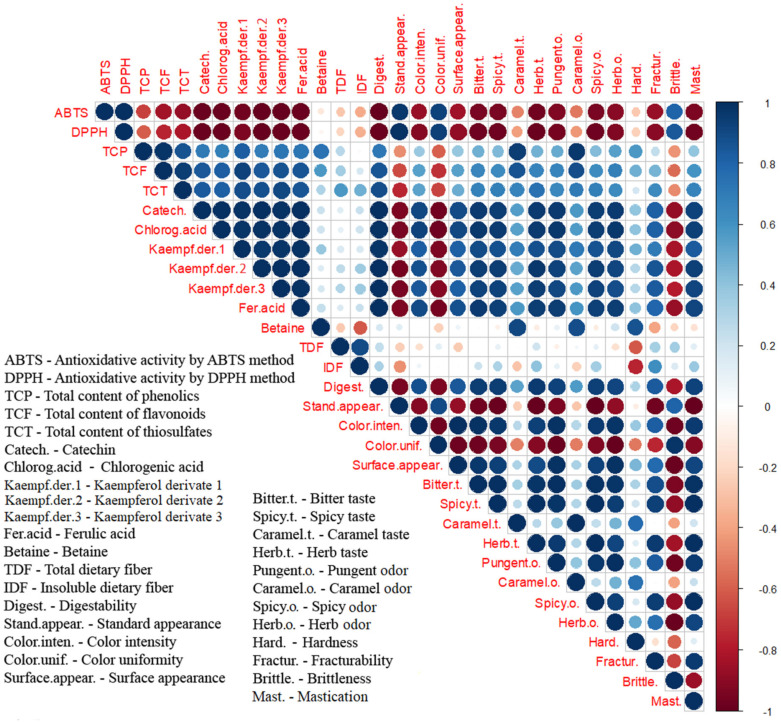
Color correlation diagram between 31 tested responses of cookies, with and without the addition of fresh and osmodehydrated wild garlic.

**Figure 4 foods-13-01941-f004:**
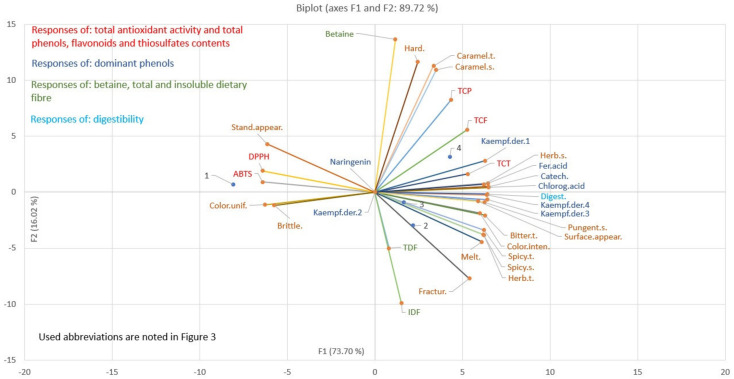
PCA of the tested cookies, with and without the addition of fresh and osmodehydrated wild garlic.

**Figure 5 foods-13-01941-f005:**
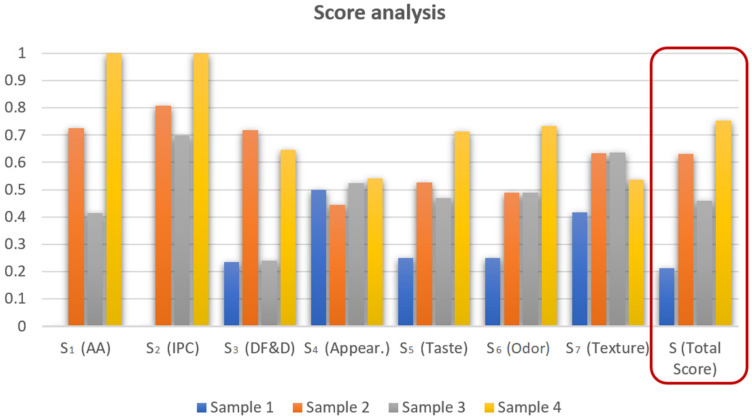
Results of the Z-score analysis of cookies with and without the addition of fresh and dehydrated wild garlic. AA—Antioxidative activity, IPC—Individual phenolic compounds, DF&D—Dietary fibers and digestibility, Appear.—Descriptive sensory analysis of appearance, Taste—Descriptive sensory analysis of taste, Odor—Descriptive sensory analysis of odor, Texture—Descriptive sensory analysis of texture.

**Table 1 foods-13-01941-t001:** Formula for tested cookie samples.

Ingredient	Sample Number			
	1	2	3	4
Quantity of wild garlic (%d.m. on flour d.m.)	0	2.50	2.50	2.50
Spelt flour (g)	193.50	188.67	188.67	188.67
Margarine (g)	64.00	64.00	64.00	64.00
Sugar (g)	2.25	2.25	2.25	2.25
NaCl (g)	2.10	2.10	2.10	2.10
NaHCO_3_ (g)	2.50	2.50	2.50	2.50
Distilled water (g)	50.00	0.00	38.8	46.3
Oregano (g)	1.93	1.93	1.93	1.93
Wild garlic (g)	0.00	55.00	16.00	8.50

**Table 2 foods-13-01941-t002:** The results of total antioxidant activity and total phenols, flavonoids, and thiosulfates contents in the tested cookie samples.

Parameter	Sample Number			
	1	2	3	4
Total content of phenolics (g GAE/100 g d.m.)	0.24 ± 0.01 ^a^	0.30 ± 0.00 ^b^	0.25 ± 0.01 ^a^	0.42 ± 0.01 ^c^
Total content of flavonoids (g CE/100 g d.m.)	0.09 ± 0.01 ^a^	0.16 ± 0.00 ^c^	0.12 ± 0.01 ^b^	0.22 ± 0.01 ^d^
Total content of thiosulfates (g AE/100 g d.m.)	0.25 ± 0.03 ^a^	0.36 ± 0.02 ^b^	0.28 ± 0.01 ^a^	0.38 ± 0.04 ^b^
ABTS IC_50_ (mg/mL)	12.65 ± 0.80 ^a^	12.39 ± 0.60 ^a^	12.45 ± 0.70 ^a^	12.37 ± 0.60 ^a^
DPPH IC_50_ (mg/mL)	64.09 ± 1.50 ^a^	62.82 ± 1.20 ^a^	63.01 ± 1.10 ^a^	62.80 ± 1.30 ^a^

^a–d^ Different letters in superscript of the same table row indicate a statistically significant difference between values at the level of significance of *p* < 0.05 (based on post hoc Tukey HSD test). GAE—gallic acid equivalent, CE—catechin equivalent, AE—allicin equivalents.

**Table 3 foods-13-01941-t003:** Concentration of dominant phenols in the extracts of the tested cookie samples.

	Sample Number			
	1	2	3	4
Detected compound:	Concentration (mg/100 g d.m.)			
Catechin	28.34 ± 0.93 ^a^	57.48 ± 1.94 ^b^	55.39 ± 2.02 ^b^	64.75 ± 3.42 ^c^
Chlorogenic acid	n.d.	10.25 ± 0.34 ^a^	10.02 ± 0.41 ^a^	12.89 ± 0.67 ^b^
Kaempferol derivative 1	n.d.	35.42 ± 1.03 ^b^	30.93 ± 1.19 ^a^	52.94 ± 2.21 ^c^
Kaempferol derivative 2	n.d.	9.37 ± 0.58 ^b^	7.37 ± 0.41 ^a^	10.34 ± 0.39 ^b^
Kaempferol derivative 3	n.d.	8.28 ± 0.34 ^b^	6.09 ± 0.21 ^a^	9.32 ± 0.40 ^c^
Ferulic acid	n.d.	11.04 ± 0.76 ^a^	10.03 ± 0.61 ^a^	14.07 ± 0.49 ^b^

^a–c^ Different letters in superscript of the same table row indicate a statistically significant difference between values at the level of significance of *p* < 0.05 (based on post hoc Tukey HSD test). n.d.: Not Detected.

**Table 4 foods-13-01941-t004:** Content of betaine, total and insoluble dietary fiber in the tested samples.

Sample Number	Betaine Content (mg/100 g d.m.)	Total Dietary Fiber (%)	Insoluble Dietary Fiber (%)
1	538.79 ± 2.24 ^b^	13.66 ± 0.17 ^b^	4.46 ± 0.03 ^a^
2	485.42 ± 4.01 ^a^	17.03 ± 0.09 ^d^	6.96 ± 0.08 ^c^
3	489.51 ± 3.99 ^a^	11.22 ± 0.10 ^a^	4.20 ± 0.02 ^b^
4	870.02 ± 4.81 ^c^	14.04 ± 0.14 ^c^	4.47 ± 0.05 ^a^

^a–d^ Different letters in superscript of the same table column indicate a statistically significant difference between values at the level of significance of *p* < 0.05 (based on post hoc Tukey HSD test).

**Table 5 foods-13-01941-t005:** Results of a descriptive sensory analysis of cookies with fresh and osmotic dehydrated wild garlic leaves.

Sample Number	Appearance				Taste			
	Standard	Color Intensity	Color Uniformity	Surface Appearance	Bitter	Spicy	Caramel	Herb
1	8.00 ± 0.17 ^a^	5.33 ± 0.33 ^ab^	9.00 ± 0.00 ^a^	4.25 ± 0.25 ^a^	1.00 ± 0.00 ^a^	1.00 ± 0.00 ^a^	2.00 ± 0.00 ^a^	1.00 ± 0.00 ^a^
2	7.00 ± 0.00 ^c^	7.25 ± 0.25 ^ef^	5.50 ± 0.50 ^ef^	6.00 ± 0.17 ^d^	6.83 ± 0.17 ^hi^	8.00 ± 0.00 ^h^	2.00 ± 0.00 ^a^	9.00 ± 0.00 ^i^
3	7.17 ± 0.17 ^c^	8.00 ± 0.00 ^gh^	4.00 ± 0.25 ^gh^	7.00 ± 0.00 ^ef^	7.50 ± 0.25 ^j^	8.00 ± 0.00 ^h^	2.00 ± 0.00 ^a^	8.00 ± 0.00 ^h^
4	7.17 ± 0.17 ^c^	7.50 ± 0.00 ^fg^	3.50 ± 0.25 ^h^	6.50 ± 0.00 ^de^	7.17 ± 0.17 ^ij^	7.50 ± 0.00 ^gh^	5.00 ± 0.00 ^c^	8.00 ± 0.00 ^h^
	**Odor**				**Texture**			
	**Pungent**	**Caramel**	**Spicy herb**	**Herb**	**Hardness**	**Fracturability**	**Brittleness**	**Mastication**
1	1.00 ± 0.00 ^a^	2.00 ± 0.00 ^a^	1.00 ± 0.00 ^a^	1.00 ± 0.00 ^a^	6.50 ± 0.50 ^a–c^	3.33 ± 0.33 ^ab^	9.00 ± 0.00 ^a^	3.00 ± 0.00 ^a^
2	6.00 ± 0.00 ^g–i^	2.17 ± 0.17 ^a^	7.17 ± 0.17 ^hi^	6.17 ± 0.17 ^f^	6.00 ± 0.00 ^a^	6.25 ± 0.25 ^gh^	6.87 ± 0.17 ^b^	7.17 ± 0.17 ^d^
3	7.25 ± 0.25 ^j^	2.00 ± 0.00 ^a^	7.00 ± 0.17 ^hi^	8.25 ± 0.25 ^h^	7.00 ± 0.00 ^cd^	5.50 ± 0.25 ^e–g^	5.00 ± 0.17 ^de^	7.00 ± 0.00 ^cd^
4	7.00 ± 0.25 ^j^	5.33 ± 0.33 ^c^	6.50 ± 0.00 ^gf^	8.33 ± 0.33 ^h^	7.50 ± 0.00 ^de^	5.00 ± 0.00 ^d-f^	5.50 ± 0.25 ^cd^	6.50 ± 0.25 ^bc^

^a–j^ Different letters in superscript of the same table column indicate a statistically significant difference between values at level of significance of *p* < 0.05 (based on post hoc Tukey HSD test).

## Data Availability

The original contributions presented in the study are included in the article/Appendix A, further inquiries can be directed to the corresponding author.

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
