# Peer review of "Evaluation of Cookies Enriched with Osmodehydrated Wild Garlic from Nutritional and Sensory Aspects"

_foods, 2024, doi:10.3390/foods13121941_

Round 1

Reviewer 1 Report

Comments and Suggestions for Authors

The topic of using wild garlic leaves as a functional ingredient in cookies sounds even interesting, however, in my opinion first raw material should be characterised and then after addition to the product. Also, the quality of the prepared manuscript is not good. Generally, it's chaotically prepared. 

At first, the abstract needs some numerical findings. The majority of the abstract text is an enumeration of methods that were used. No place for findings and presentation of any relation found in statistics.

Methods are well described, however, some of them as spectrophotometrical ones are already well-known e.g. TPC, and TF (these abbreviations are used), and in my opinion, too much attention is focused on methodology rather than on the description and discussion. Either 2.2 Osmotic dehydration procedure, or 2.8 In vitro digestion, or could be presented as a schema. All of the methods should be shortened and only contain references and main parameters that were modified. 2.10 Methods of statistical analysis should be moved to "supplementary materials".

No chemicals are listed at the beginning of the section "materials and methods".

Line 484: Are authors sure that this is a "column"?

Instead of a sample number write a code and below write the meaning of numbers or abbreviations. 

Line 506-511: it is about wild garlic itself, therefore such identification should be done for garlic used in this study.

I don't also see the point of the correlation of texture with DPPH and ABTS. 

Comments on the Quality of English Language

The manuscript required an English editing. 

Author Response

The responses to Reviewer 1 are presented in the pdf file

Reviewer 2 Report

Comments and Suggestions for Authors

This is a very detailed study that analyzes the nutritional and sensory properties of spelt cookie with wild garlic. However, in the part of the result and discussion, many of the phenomena found failed to make reasonable inferences and find out the information in the references to support them. Also, in the section on sensory evalution, the inferences drawn from sensory characteristics such as odor and taste are not very reasonable. It is suggested that all of these should be further revised. 

#1. In vitro digestion component of the Materials and Methods, Use of AA% as an indicator for assessment, please add the references.

#2. In the section of 2.10.4, the response values mathematically transformed by min-max normalization. But the segment Z-score showed in 379-424 do not list the correct references, also the reference no. 7 in line 374 does not use Z Score. 

#3. Detailed methods of sensory evaluation in 2.9 are not clearly described, e.g. mouth rinsing, sample arrangement, the method of biscuit eating, etc.

#4. The SD of the antioxidant indicators in Table 2 is very low and not normal. The IC50 is calculated by conversion, is it because the SD is not converted during the conversion process? So please list the calculation method of SD.

#5. SD is not listed in Table 3.

#6. In Line 530-532, it showed that “chlorogenic acid was identified for the first time by the HPLC analysis carried out within the presented research”. But in Line542-545, it showed “sugar beet molasses contains 36.82 mg/100g d.m. chlorogenic acid”. This result shows that the source of chlorogenic acid should not be only from sugar beet molasses. Please provide a reasonable explanation of the origin of chlorogenic acid and list the references.

#7. In Line 554-555, it showed that “aim of increasing the level of betaine in these products”. Please explain why only betaine is being discussed.

#8. In Line 561-562, it showed that “betaine of 561 mg/100g was detected, which is probably fully contributed by whole spelt flour”. Compared with the results in Table 1, the amount of spelt flour used in Samples 2-4 was very similar, but why was there a decrease in contents of betaine. Please provide a reasonable explanation and list the references.

#9. In Line 3572-573, it showed that “A higher addition of osmodehydrated wild garlic in molasses, which is possible up to 30%, would increase the betaine content by about 18 times”. However, the article does not mention how the maximum dosage of 30% was obtained. Have you actually conducted the relevant experiments? Please explain why the 30% was obtained.

#10. In Line 660-662, it showed that “Therefore, the enrichment of cookies with wild garlic, and foremost wild garlic osmodehydrated in molasses, can increase the antioxidant activity of the product during its consumption”. However, as seen in Fig. 2, the antioxidant effect is greater for both Samples 2 and 3 than for Sample 4. Please provide a reasonable explanation and list the references.

#11. In Table 5, Color Uniformity has a score of 10, which does not meet the settings of the rating scale of 1-9.

#12. The low SD in Table 5 is not consistent with the results of the general descriptive analyses. Please provide a reasonable explanation. And the number of tastings carried out by the panelists was also not specified. 

#13. Sample 4 in Table 5 has a strong caramel taste and caramel odor, and the caramel flavour can be interpreted as a positive sensory characteristic. But the Spicy and Herb in taste, and the Pungent, Spicy , and herb in odor, it seems that these sensory characteristics of biscuits are supposed to create negative consumer perceptions. However, in the article, it is mentioned that cookies treated with wild garlic have better sensory characteristics. Please double-check this section and re-make this part of the discussion.

#14. In Fig. 4, the colours of the PCA results for each type of characteristic are very confusing, and it is recommended that different types of characteristics can be marked with different shades of colours.

#15. In Line 806-808, it showed that “the highest segment score values, significantly higher than other samples, for antioxidative activity, individual phenolic compounds, taste and odor.” As suggested earlier in point 13, these sensory characteristics of Sample 4 should be judged as negative sensory characteristics, unlike those defined in the article. Please double-check this section and re-make this part of the discussion.

#16. In countries where sensory evaluation trials are conducted, it is necessary to ask the panelists to fill in an Informed Consent Statement, if not to apply for a simple Institutional Review Board (IRB). Have the panelists been asked to fill in the Informed Consent Statement when conducting the tastings?

Comments on the Quality of English Language

Minor English editing is required

Author Response

The responses to Reviewer 2 are presented in the pdf file

Reviewer 3 Report

Comments and Suggestions for Authors

It is recommended to supplement the data for functional composition analysis of wild garlic leaves after different treatments (fresh, osmodehydrated in ternary solution and osmodehydrated in ternary solution).

Author Response

The responses to Reviewer 3 are presented in the pdf file

Round 2

Reviewer 2 Report

Comments and Suggestions for Authors

#2 In the sensory evaluation method, please check whether there are any oral cleansing steps in the evaluation of different samples. Please also indicate the experimental design method used for the randomized arrangement.

#8 The quantities of flour used in samples No. 1-3 were 193.50, 188.67 and 188.67 g respectively. However, the amount of betain has been reduced from 540 to 315 mg/100, a reduction that is very different from that of spelt flour.

#12 We understand that the analyses are experienced analyses. Please add the ANOVA table between analyses, repetitions, and samples to demonstrate that there is no longer a significant difference between trained analyses.

#13 Spicy and Herb in taste are defined as positive sensory properties, which may occur for consumers in different regions, but please provide references to substantiate your assumptions.

#16 There is no relative document provided as supplementary material in Appendix 2.

Comments on the Quality of English Language

English language fine.

Author Response

Dear Reviewer, 

the comments to the round 2 revision are presented in the pdf file, starting at the page 8 (responses to the comments of round 1 revision are presented at the first 7 pages of the document)
